# DO EXPLANATIONS GENERALIZE ACROSS LARGE REASONING MODELS?

## ABSTRACT

Large reasoning models (LRMs) produce a textual chain of thought (CoT) in the process of solving a problem. This CoT is potentially a powerful tool to understand the problem, surfacing a human-readable, natural-language explanation. However, it is unclear whether these explanations *generalize*, i.e. whether they capture general patterns about the underlying problem rather than patterns which are esoteric to the LRM. This is a crucial question in understanding or discovering new concepts, e.g. in AI for science. We study this generalization question by evaluating a specific notion of generalizability: whether explanations produced by one LRM induce the same behavior when given to other LRMs.

We find that CoT explanations do exhibit this form of generalization (i.e. they increase consistency between LRMs) and that this increased generalization is correlated with human preference rankings. We further analyze the conditions under which explanations do or do not yield consistent answers and propose a straightforward, sentence-level ensembling strategy that improves consistency. These results prescribe caution when using LRM explanations to yield new insights and outline a framework for characterizing LRM explanation generalization.

## 1 INTRODUCTION

The chains of thought (CoT) produced by large reasoning models (LRMs) have enabled strong performance on a range of complex tasks (Guo et al., 2025; Guha et al., 2025; Liu et al., 2025; Abdin et al., 2025; Agarwal et al., 2025). These CoT reasoning traces are often presented as human-readable explanations, but many researchers have questioned whether these traces can be made faithful to the true decision-making processes followed by LRMs (Barez et al., 2025; Chen et al., 2024; Shojaee et al., 2025; Xiong et al., 2025). In this paper, we examine a different issue that is related to faithfulness: we investigate the generalization of reasoning traces across different LRMs.

Our inquiry is motivated by the search for good natural-language explanations. For it is not good enough for an explanation to be correct; it is also important for an explanation to be *learnable*. That means: if we give the explanation to another person (or another agent), they should be able to understand it and draw the intended conclusions from it. In our setting, we quantify this question in terms of cross-LRM CoT generalization.

Importantly, models and humans may provide different explanations for the same concept, and uniformity is not our goal. Instead, we ask whether a given explanation, once produced, reliably guides other models to the same answer. This perspective reveals an underexplored dimension of CoT research: is a reasoning CoT produced by one LRM generalizable enough that it can lead a different LRM to follow the same reasoning, producing the same answer? Posing the question in this way enables an automated, quantitative evaluation of generalization for explanations, which has remained elusive despite generalization being a cornerstone of statistical machine learning. Practically, this question is critical for understanding whether an LRM explanation can provide usable insights into how a problem is solved. This question is especially critical in scientific discovery, where explanations that capture problem-level patterns, rather than model-specific quirks, could inspire novel human insights (Schut et al., 2025; Singh et al., 2024), especially as LRMs reach superhuman capabilities in domains such as science and mathematics (Wang et al., 2023; Romera-Paredes et al., 2024) and are increasingly used in educational settings (Kasneci et al., 2023; Bewersdorff et al., 2025).

We evaluate the effect of LRM explanations on improving the consistency between LRM answers in different ways (Fig. 1) across MedCalc-Bench. We find that LRM explanations do generalize, i.e. they increase consistency between LRMs (Fig. 1E), even improving consistency when the underlying explanation suggests a wrong answer. To further improve generalization, we propose a straightforward, sentence-level ensembling strategy that encourages the production of explanations less tied to the idiosyncrasies of any single model; we find that this further increases consistency between LRMs (Fig. 1E).

To evaluate the relationship between cross-model consistency and human preferences for explanations, we conduct a human study evaluating human preferences for various CoT explanations, suggesting that more consistent explanations may also be preferable for human users. Together, these results represent a step towards eliciting explanations from LRMs that are both transferable across models and informative to human users.

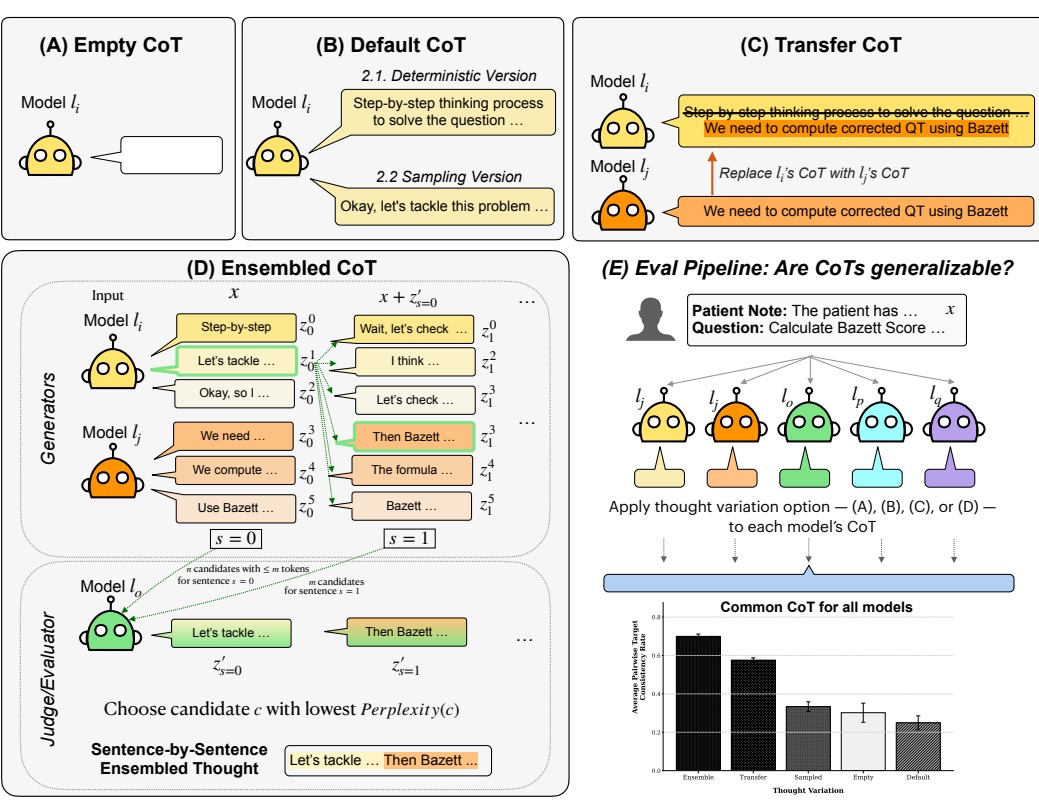

Figure 1: *Methods for eliciting reasoning chains and exploring generalization of Chain-of-Thought (CoT).* The figure illustrates four approaches to modifying or replacing model-generated CoTs. **Panel (E)** (bottom-right) shows how CoT generalization is evaluated for the MedCalc-Bench dataset across a set of models (A–E), where each model's reasoning can be substituted with one of the following variations: **(A) Empty CoT:** No reasoning text is provided between the model's thinking tags. **(B) Default CoT:** The model's own reasoning is used. (2.1) uses deterministic decoding, while (2.2) uses nucleus sampling (`do_sampling=True`). **(C) Transfer CoT:** Reasoning from one model is directly transferred to another, replacing its own. **(D) Ensembled CoT:** A generator–evaluator loop. Generator models produce $n = 3$ candidate sentences ($\leq 15$ tokens each), forming $k$ candidates. These are scored by the evaluator, and the least surprising candidate (lowest perplexity) is appended to the growing ensembled thought. This updated context is fed back into the generators, and the process repeats until an end-of-thought or maximum token limit is reached.

## 2 METHODS

**Evaluating explanation generalizability**   Building on prior works that have focused on evaluating the faithfulness of an LRM explanation to a single model's reasoning, we evaluate the generalization of an explanation to a new LRM. Intuitively, a new LRM (trained using different techniques / datasets), should be a stand-in for a user simulator, allowing for evaluating whether an explanation is generalizable. Note, however, that this assumption may fail in the case that different LRMs share a common bias for a particular explanation.

Fig. 1 gives an overview of the different methods for eliciting and using LRM explanations that we consider. Concretely, given an LRM $l_{\text{gen}}$ and a problem string $x$, we elicit a reasoning explanation by passing a model-specific prompt, that elicits a reasoning explanation $z = l_{\text{gen}}(x)$ before producing an answer $a = l_{\text{gen}}(x|z)$. For example, with the `Qwen/QwQ-32B` model (Team, 2025), we use a prompt of the form: `{Problem}<think>{Thinking Text}</think>{Answer}`....

When we test generalization, we supply the explanation from $l_{\text{gen}}$, i.e., $z = l_{\text{gen}}(x)$ to a different LRM $l_{\text{eval}}$ amongst the population of models $L = \{l_i, l_j...l_q\}$. We then produce an answer given an LRM $l_{\text{eval}}$, by giving it the explanation within the think tags. These explanations often contain answers. To ensure that the answer is not directly contained in the explanation, we further process the explanation by removing explicit answer declarations detected by an LLM.

We measure two metrics: accuracy $A$ (measured via ground truth correctness) and consistency $G$ (quantified by the frequency of matching responses). Concretely,

$$A = I(l_{\text{eval}}(x|l_{\text{gen}}(x)), a), \qquad G = \sum_{\substack{l \in L \\ z = l_{\text{gen}}(x)}} I(l_{\text{eval}}(x|z), l(x|z)) \tag{1}$$

where $I$ is the scoring function specific to a dataset to see if two answers match.

**Eliciting ensemble explanations**   Apart from extracting chain-of-thoughts from $l_{\text{gen}}$ with various settings (empty, default, sampling), we also generate a chain-of-thought from a set of $l_{\text{gen}}$s, i.e, $L_{\text{gen}} = \{l_i, l_j, l_o...\}$. Given this set of LRMs, we designate a subset as generators and a separate model as the evaluator. At each step, the generators produce $n$ candidate sentences with $m$ tokens ($n = 3$ and $m = 15$, in our case) conditioned on the context, which consists of the problem string $x$. The evaluator then selects the candidate with the lowest perplexity, which is appended to the ensembled chain of thought. The context is subsequently updated to include the original problem and all accumulated ensembled sentences. This sentence is part of $z$, which would be of size $s$, where size indicates the number of sentences we generate to create a complete ensembled thought. This process repeats until one of the generator models outputs an end-of-thought token or the maximum chain length ($m \cdot s$) is reached.

**CoT variations**   As illustrated in Fig. 1, we evaluate four variations of CoT generation across models.

1. Empty CoT: The think text is an empty string, serving as a baseline method. Therefore, when the model generates its final answer, the preceding context is `{Problem}<starting-think-tag>""<closing-think-tag>`

2. Default CoT: The standard setting used in prior benchmarks, where the think text is generated by the model without modification. This method includes two sub-variations: one without sampling and one with sampling, covering both deterministic and non-deterministic default behaviors.

3. Transfer CoT: The think text is replaced by a default deterministic think text of another model. We test on various permutations of the models to see how different models' reasoning traces generalize across other models.

4. Ensembled Thoughts: The think text is replaced by explanations generated via Ensemble explanations.

For each thought type, we also present two versions — (1) with complete text, (2) with just hints and explanations, i.e., without answers. The intention behind this versioning is to understand to what

Table 1: LRMs used in this study. We focus on recent LRMs that have shown to be capable in various reasoning tasks.

| Alias | Model | Huggingface ID | Citation |
|---|---|---|---|
| NRR | Nemotron-Research-Reasoning-Qwen-1.5B | `nvidia/Nemotron-Research-Reasoning-Qwen-1.5B` | (Liu et al., 2025) |
| OpenT | OpenThinker-7B | `open-thoughts/OpenThinker-7B` | (Guha et al., 2025) |
| OSS | gpt-oss-20b | `openai/gpt-oss-20b` | (OpenAI, 2025) |
| QwQ | Qwen/QwQ-32B | `Qwen/QwQ-32B` | (Team, 2025) |
| DAPO | DAPO-Qwen-32B | `BytedTsinghua-SIA/DAPO-Qwen-32B` | (Yu et al., 2025) |

extent answers in chain of thoughts affect generalizability and accuracy. To remove answers, we use OpenAI's o4-mini model (OpenAI, 2025) as detailed in Appendix C.

## 3 RESULTS

### 3.1 EXPERIMENTAL SETUP

**Datasets** To assess reasoning in a specialized and general domains, we adopt two benchmarks: We use `MedCalc-Bench` (Khandekar et al., 2024) to target medical domain-specific reasoning, and `Instruction Induction` (Honovich et al., 2022), which evaluates general reasoning capabilities. We extend the latter benchmark by incorporating 12 additional tasks to capture more complex general reasoning.

1. `MedCalc-Bench` (Khandekar et al., 2024): Each instance consists of a patient note and a question, which asks to compute a specific clinical value. We have a randomly chosen representation sample of size 100 of such calculation tasks.

2. `Instruction Induction` (Honovich et al., 2022): Each instance presents five input-output pairs and the model is tasked to generate a natural language instruction that captures their underlying relation. We evaluate this benchmark using 100 samples drawn from 20 tasks, with five examples per task.

In Eq. (1), the scoring function, $I$, for `MedCalc-Bench` is exact-matching while for `Instruction Induction` is BERTScore Zhang et al. (2020).

**Models** We deploy a series of LRMs; each listed in Table 1. We use huggingface implementations (Wolf et al., 2020) of each model in `bfloat16`.

**User study** We designed and conducted a user study with 15 participants to investigate whether greater generalizability correlates with users' perceptions of model CoT quality. The study was administered via Qualtrics, where participants received an anonymous survey link. Respondents, who were computer science and healthcare researchers, were instructed to evaluate reasoning chains from several model variations (kept anonymous to participants) across the following criteria: Clarity of Steps, Ease of Following, Confidence, followed by a Best Overall ranking. The questions were posed in Likert-scale manner:

- *Clarity of Steps:* The reasoning steps were clear and well explained (1 = Very unclear; 5 = Very clear)

- *Ease of Following*: The answer follows clearly from the reasoning steps. (1 = Very difficult; 5 = Very confident)

Table 2: Model configurations and reasoning approaches evaluated in the user study.

| Reasoning Approach | Model Configuration |
| --- | --- |
| Deterministic Chain-of-Thought (CoT) | GPT-OSS-20B |
| Deterministic Chain-of-Thought (CoT) | DAPO-Qwen-32B |
| Ensemble CoT (Generator / Evaluator) | QwQ-32B + DAPO-Qwen-32B / GPT-OSS-20B |
| Ensemble CoT (Generator / Evaluator) | QwQ-32B + GPT-OSS-20B / DAPO-Qwen-32B |

- *Confidence:* After reading, I feel confident I understood the reasoning. (1 = Not confident at all; 5 = Very confident)

The *Best Overall* ranking was asked in the following way: "Rank the following models' Chain-of-Thought explanations from most understandable to least understandable" (1 is the most understandable). While conducting this study, we did not collect demographic or any other personal identifying information. Because it would be unreasonable to expect participants to perform domain-specific tasks, our human evaluation focuses on whether the explanations are convincing or helpful, in contrast to the model-oriented accuracy and consistency metrics. All chain-of-thoughts were generated using `MedCalc-Bench` and the models and thought variations are summarized in Table 2. We constructed 10 examples, each paired with 4 chain-of-thoughts. For evaluation, participants were shown shown 5 examples, randomly selected and balanced across conditions.

## 3.2 EVALUATING GENERALIZABILITY OF LRM CoT EXPLANATIONS

> **Generalization without accuracy**
>
> LRM explanations generalize, even when the explanation induces an inaccurate answer.

Fig. 1 shows that the consistency rate among the models' conclusions increases in both transferred and ensemble settings. This includes 'incorrect' answers, where models arrive at the same wrong conclusion when provided with identical chains-of-thought. In Fig. 2, we measure how often a CoT leads models to converge on the same answer even when the predicted answer is not mentioned in the CoT (i.e., when the CoT mostly provides partial hints or reasoning). The left panel breaks down these "same answer" cases into convergence on the same correct versus the same incorrect answers. The right panel shows the proportion of consistently incorrect answers across CoT types. These results demonstrate that CoTs can systematically steer model reasoning, including toward the same conclusion, which indicates that CoT can exert a generalizing influence on model behavior even when the reasoning they provide can be incorrect. In Fig. 3, we report a breakdown of outcomes comparing CoT and baseline empty CoT reasoning across several scenarios, including cases where (a) an incorrect model prediction becomes correct after CoT transfer, (b) a correct prediction is perturbed, and (c) different forms of agreement or disagreement emerge across models. Regarding accuracy, Table 3 shows the effect of these chains-of-thought on different models. The takeaway is that models with weaker baseline accuracy can reduce the accuracy of other models when their CoTs are transferred even if the other model inherently performs better at their own baseline.

## 3.3 USER STUDY

> **Human explanation preferences**
>
> LRM explanations that are more consistent receive higher user preference ratings.

Fig. 11 presents the statistical overview of our user study, showing box plots for each model and criterion. The plots display the mean (red diamond), median (red line), interquartile range (box boundaries), whiskers, and outliers (points). Overall, the default DAPO CoT and the ensembled QwQ+DAPO/OSS CoT were consistently perceived easier to understand than the default OSS CoT and the esembled QwQ+OSS/DAPO CoT.

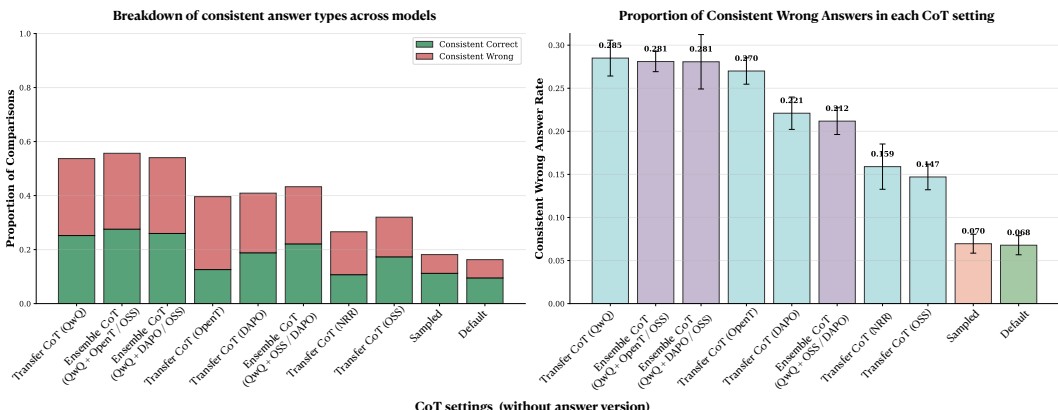

Figure 2: **Consistency breakdown across thought variations without answers.** *Left:* Proportion of consistent outputs separated into matching correct and matching incorrect conclusions. *Right:* Rate of consistent answers that are wrong across various thought settings.

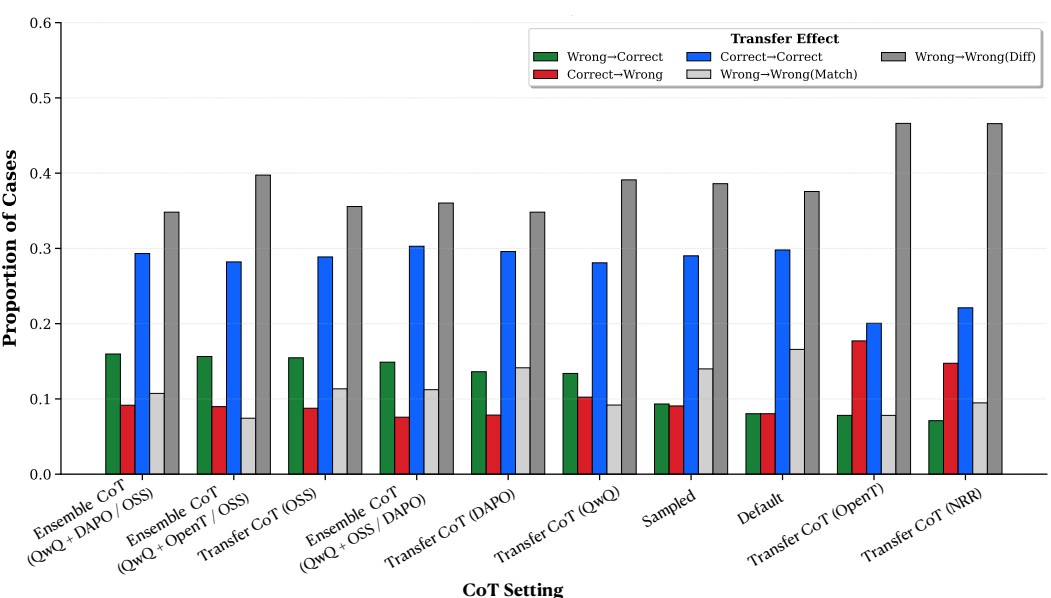

Figure 3: **CoT Transfer Effect Analysis** Distribution of transfer outcomes when Chain-of-Thought (CoT) reasoning is used or transferred across models. Each CoT setting is evaluated by comparing model predictions with CoT (which have explanations with answers removed) versus without CoT (empty baseline). Settings include, Default: models using their own generated CoT; Sampled: CoT generated through sampling; Transfer CoT $l_{gen}$: CoT transferred from model $l_{gen}$ to all models; Ensemble CoT: combined CoT from multiple models. The five conditions represent: Wrong→Correct: cases where CoT successfully corrects errors (green); Correct→Wrong: cases where CoT misleads the model from correct to incorrect predictions (red); Correct→Correct: cases where CoT maintains correct predictions (blue); Wrong→Wrong(Match): both predictions incorrect with identical wrong answers (light gray); Wrong→Wrong(Diff): both predictions incorrect with different wrong answers (dark gray). Results are aggregated across multiple target models for each CoT setting. Settings are sorted by Wrong→Correct rate (descending).

Independent t-tests with Bonferroni correction confirmed that OSS was rated significantly worse than both DAPO ($p < 0.0001$) and QwQ+DAPO/OSS ($p < 0.0001$) in terms of *Clarity of Steps*. This pattern extended to *Ease of Following* and *Confidence*. In contrast, one ensemble variant per-

Table 3: Comparison of CoT accuracy across models for MedCalc-Bench and Instruction Induction

| Method | Setting | MedCalc-Bench (Exact-Match) | | | | | | Instruction Induction (BERTScore) | | | | | |
| | | NRR 1.5B | OpenT 7B | OSS 20B | QwQ 32B | DAPO 32B | Avg | NRR 1.5B | OpenT 7B | OSS 20B | QwQ 32B | DAPO 32B | Avg |
|---|---|---|---|---|---|---|---|---|---|---|---|---|---|
| Empty CoT | No text | 0.10 | 0.18 | 0.45 | 0.36 | 0.38 | 0.29 | 0.53 | 0.55 | 0.56 | 0.55 | 0.57 | 0.55 |
| Default CoT | Full text | 0.13 | 0.24 | 0.43 | 0.41 | 0.41 | 0.32 | 0.58 | 0.56 | 0.61 | 0.61 | 0.62 | 0.60 |
| | W/o Ans | 0.14 | 0.24 | 0.43 | 0.38 | 0.41 | 0.32 | 0.58 | 0.46 | 0.61 | 0.60 | 0.62 | 0.57 |
| Sampled CoT | Full text | 0.14 | 0.32 | 0.47 | 0.39 | 0.37 | 0.34 | 0.58 | 0.50 | 0.52 | 0.57 | 0.60 | 0.55 |
| | W/o Ans | 0.16 | 0.29 | 0.45 | 0.38 | 0.35 | 0.33 | 0.56 | 0.56 | 0.60 | 0.60 | 0.60 | 0.58 |
| Trans. CoT (NRR) | Full text | 0.13 | 0.13 | 0.21 | 0.22 | 0.29 | 0.20 | 0.58 | 0.56 | 0.60 | 0.58 | 0.62 | 0.59 |
| | W/o Ans | 0.14 | 0.15 | 0.24 | 0.30 | 0.34 | 0.23 | 0.58 | 0.57 | 0.60 | 0.60 | 0.63 | 0.60 |
| Trans. CoT (OpenT) | Full text | 0.24 | 0.26 | 0.24 | 0.26 | 0.27 | 0.25 | 0.60 | 0.57 | 0.43 | 0.62 | 0.62 | 0.57 |
| | W/o Ans | 0.21 | 0.24 | 0.26 | 0.26 | 0.25 | 0.24 | 0.60 | 0.46 | 0.59 | 0.59 | 0.61 | 0.57 |
| Trans. CoT (OSS) | Full text | 0.39 | 0.40 | 0.43 | 0.42 | 0.39 | 0.41 | 0.60 | 0.57 | 0.61 | 0.61 | 0.61 | 0.60 |
| | W/o Ans | 0.26 | 0.44 | 0.43 | 0.40 | 0.44 | 0.39 | 0.60 | 0.57 | 0.61 | 0.60 | 0.62 | 0.60 |
| Trans. CoT (QwQ) | Full text | 0.41 | 0.40 | 0.40 | 0.41 | 0.41 | 0.41 | 0.61 | 0.57 | 0.52 | 0.61 | 0.62 | 0.59 |
| | W/o Ans | 0.34 | 0.37 | 0.39 | 0.38 | 0.37 | 0.37 | 0.60 | 0.57 | 0.61 | 0.60 | 0.62 | 0.60 |
| Trans. CoT (DAPO) | Full text | 0.38 | 0.40 | 0.45 | 0.40 | 0.41 | 0.41 | 0.62 | 0.58 | 0.53 | 0.62 | 0.62 | 0.60 |
| | W/o Ans | 0.31 | 0.40 | 0.39 | 0.40 | 0.41 | 0.38 | 0.60 | 0.58 | 0.62 | 0.61 | 0.62 | 0.61 |
| Ens. (QwQ+DAPO/OSS) | Full text | 0.40 | 0.39 | 0.39 | 0.39 | 0.40 | 0.39 | 0.60 | 0.57 | 0.60 | 0.61 | 0.61 | 0.60 |
| | W/o Ans | 0.39 | 0.37 | 0.41 | 0.41 | 0.40 | 0.40 | 0.60 | 0.57 | 0.61 | 0.60 | 0.62 | 0.60 |
| Ens. (QwQ+OSS/DAPO) | Full text | 0.40 | 0.39 | 0.46 | 0.43 | 0.43 | 0.39 | 0.62 | 0.57 | 0.62 | 0.62 | 0.62 | 0.61 |
| | W/o Ans | 0.28 | 0.37 | 0.45 | 0.42 | 0.43 | 0.38 | 0.60 | 0.57 | 0.61 | 0.60 | 0.62 | 0.60 |
| Ens. (QwQ+OpenT/OSS) | Full text | 0.37 | 0.37 | 0.41 | 0.38 | 0.39 | 0.38 | 0.61 | 0.57 | 0.61 | 0.61 | 0.62 | 0.60 |
| | W/o Ans | 0.34 | 0.41 | 0.42 | 0.38 | 0.40 | 0.39 | 0.61 | 0.58 | 0.61 | 0.60 | 0.62 | 0.60 |

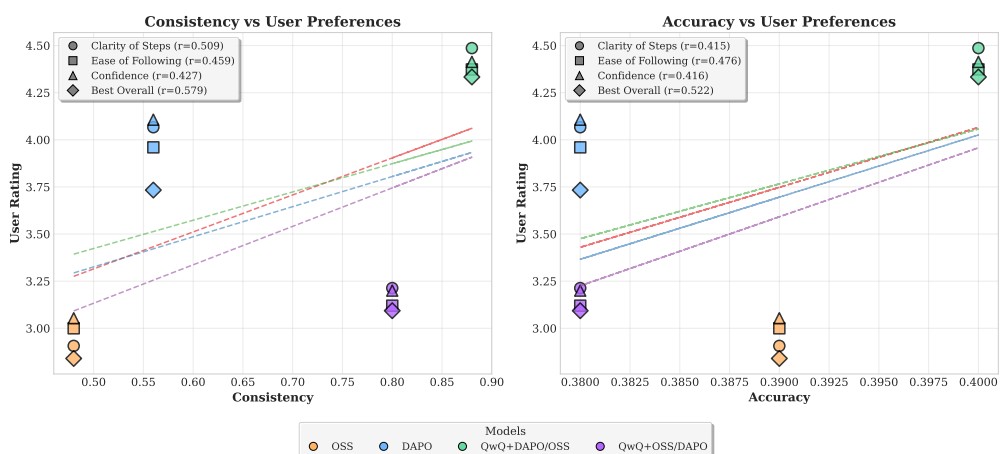

Figure 4: Scatter-and-trendline plots examining how model consistency (left) and model accuracy (right) relate to human user preferences. Each plot includes four user-rated dimensions: Clarity of Steps, Ease of Following, Confidence, and Best Overall. It also includes linear regression lines showing the strength and direction of their correlations. Consistency has a stronger and more visibly separable relationship with user preference than accuracy.

formed similarly to OSS, and the difference was not statistically significant ($p = 1.0$). All other comparison results, including those for *Best Overall*, showcased significant differences.

Between DAPO and ensembled QwQ+DAPO/OSS CoT, no significant differences emerged for *Clarity of Steps*, *Ease of Following* and *Confidence*. We also ran Wilcoxon signed-rank test and paired t-test. It yielded concordant results (see Table 4). The non-parametric and parametric tests agreed on significance for 23 out of 24 comparisons. For *Best Overall*, the ensemble was rated significantly higher ($p = 0.005$). These findings suggest that the ensemble with DAPO as generator and OSS as evaluator is the most effective configuration.

With Ensemble CoT + DAPO / OSS performing the best in Fig. 5 and in this study, the results support the claim that improving the generalizability of CoT can enhance perceived model quality explanation. In Fig. 4, we include a scatter plot illustrating this correlation. Notably, consistency appears to be a stronger predictor of user satisfaction than accuracy.

### 3.4 ANALYSIS

Fig. 5 provides a detailed analysis of the average pairwise target consistency rate across different thought variations and configurations in both benchmarks. Pairwise target consistency is defined as the average consistency between pairs of models' outputs when provided with a chain of thought. We observe a clear increase in consistency from the default, empty, and sampled groups to the ensemble and transfer settings. Among these, the strongest performance comes from specific combinations of ensemble methods and thought variations in both MedCalc-Bench and Instruction Induction. In MedCalc-Bench, ensembles that use OSS as the evaluator achieve substantially higher consistency than other configurations. When OSS serves as the generator, consistency decreases, remaining above OSS on its own but closer to the lower end of the distribution. In Instruction-Induction, transferring OSS's CoT yields the strongest performance compared to other transfers. Similarly, the ensemble that uses OSS as the generator outperforms the other ensemble configurations. Taken together, these results suggest that models whose CoT transfers exhibit greater consistency also tend to function as more effective generators within ensemble transfers.

Fig. 6 examines how the average pairwise consistency rate changes depending on whether a pair includes the source model (i.e., the model whose CoT was used) or whether both models are targets that did not contribute to the CoT. This comparison highlights the extent to which similarity in responses persists when one member of the pair is the CoT creator versus when neither model generated the original chain of thought. Fig. 7 illustrate the models' self-consistency, which ranges from 20% to about 50% for respective models in question. Notably, these self-consistency levels differ from the cross-model consistency observed when models sample their responses.

### 4 RELATED WORK

**Generating and improving natural-language explanations** A large body of work extends chain-of-thought prompting (Wei et al., 2022) by probing or refining the explanations it produces. Examples include evaluating counterfactuals introduced into the chain of thought (Gat et al., 2023), testing their robustness to mistakes introduced into the reasoning chain (Lanham et al., 2023), or using contrastive CoT to induce reliance on the reasoning chain (Chia et al., 2023). A few works seek to improve the consistency in the generations made by an LLM, either between the generation and validation of LLMs (Li et al., 2023), between LLM predictions on implications of an original question (Akyürek et al., 2024), on counterfactual inputs for an original question (Chen et al., 2025b; Shihab et al., 2025), or by more generally introducing desirable structures into reasoning traces (Sun et al.). All of these methods can be used in conjunction with Ensemble explanations.

A similar line of work has studied generating explanations directly for a problem/dataset, rather than for a single example, e.g. describing distributions in natural language (Zhong et al., 2023; Singh et al., 2023) or human-readable programs (Romera-Paredes et al., 2024; Novikov et al., 2025). These works rely on some form of external verification for explanations (e.g. restricting an explanation to be python-runnable code) rather than allowing them to be flexible. A separate line of work has studied ensemble LLM generation (Tekin et al., 2024; Chen et al., 2025c), although not at the sentence-level and not for the purpose of explanation generation.

**Assessing CoT explanations** Model-generated text explanations have shown issues with faithfulness to the underlying LLM/LRM (Turpin et al., 2023; Ye & Durrett, 2022), e.g. LLM reasoning chains have been show to be inconsistent across counterfactuals (Mancoridis et al., 2025), sensitive to minor variations (Yeo et al., 2024), the answer may not follow from the chain (Xiong et al., 2025), may not reveal the info they really rely on (Chen et al., 2025a), inconsistently learn algorithms (Shojaee et al., 2025), can succeed at reasoning with invalid intermediate tokens (Stechly et al., 2025), or be trained to use dummy intermediate tokens (Pfau et al., 2024). Additionally, humans studies suggest that users perceive the wrong narratives from reasoning chains (Levy et al., 2025) and that users to not necessarily rank accurate reasoning traces for models higher (Bhambri et al., 2025a).

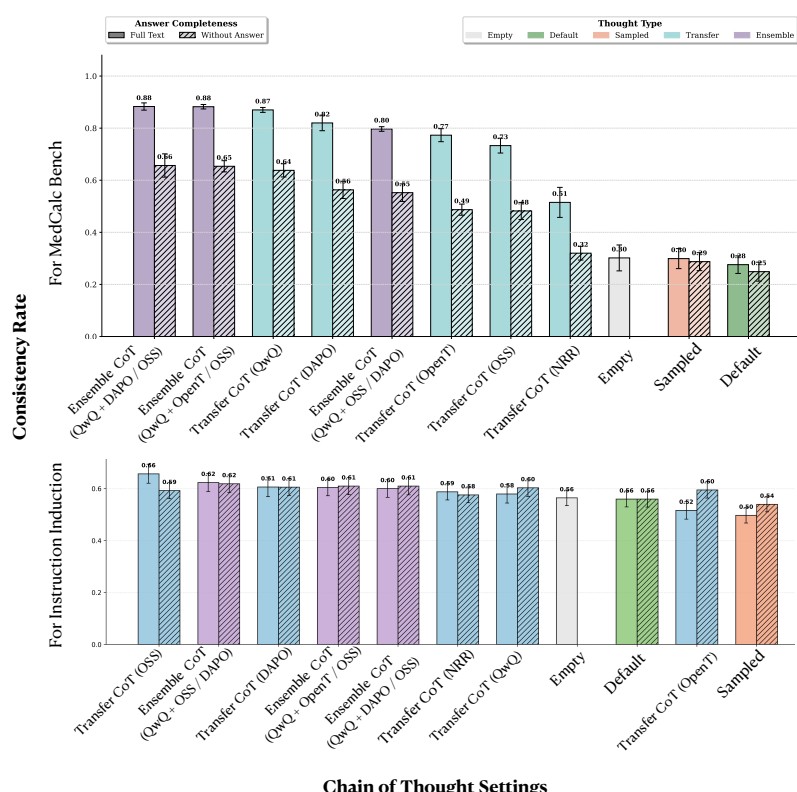

Figure 5: Average pairwise consistency across thought settings in MedCalc-Bench (above) and Instruction Induction (below). For thought variations indicating Ensemble CoT, models listed before the slash (/) serve as generators, while the model after the slash acts as the judge/evaluator. Results are reported both for the full text and for text with the final answer removed.

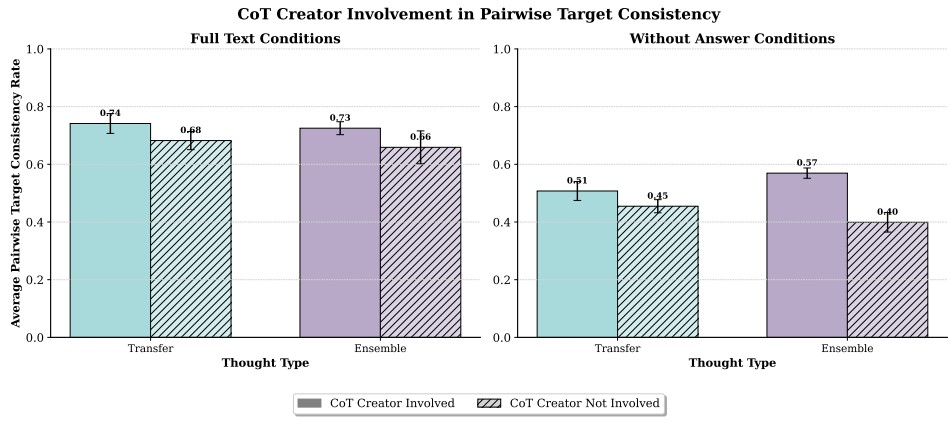

Figure 6: Average pairwise consistency for transfer and ensemble thoughts, comparing MedCalc Bench cases where a model is involved as a creator/source of the CoT versus cases where none of the models tested were part of the source.

See also other warnings about relying on LLM reasoning traces (Kambhampati et al., 2025; Bhambri et al., 2025b; Chua & Evans, 2025), including mechanistic analysis (Bogdan et al., 2025; Prakash et al., 2025), and on the difficulty of evaluating reasoning faithfulness (Zaman & Srivastava, 2025).

**Evaluating natural-language explanations** Prior works for evaluating natural-language explanations have aligned on one of three dimensions: consistency, plausibility, and faithfulness. *Consis-*

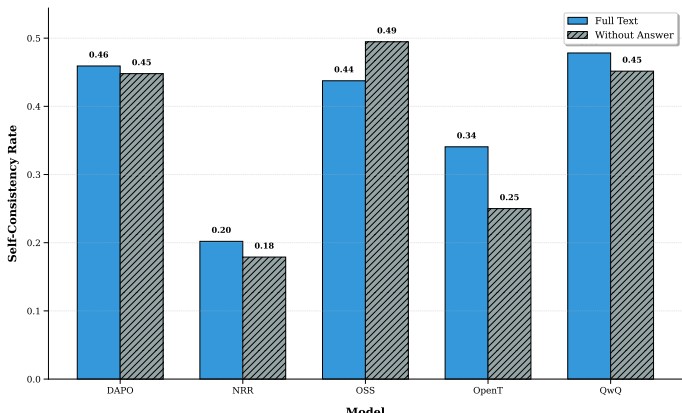

Figure 7: Model self-consistency rates for chain-of-thought generation on MedCalc-Bench. Bars compare default decoding versus sampling-based generation strategies.

*tency*, which we focus on in this work, measures if the model generates consistent explanations on similar examples (Hase & Bansal, 2020; Chen et al., 2024). *Plausibility* evaluates humans' preference of an explanation based on its factual correctness and logical coherence (Herman, 2017; Lage et al., 2019; Jacovi & Goldberg, 2020). It is different from *faithfulness*, which measures whether an explanation is consistent with the model's internal decision process (Harrington et al., 1985; Ribeiro et al., 2016; Gilpin et al., 2018; Jacovi & Goldberg, 2020). More broadly, explanation evaluation frameworks such as (Doshi-Velez & Kim, 2017), (Ribeiro et al., 2016), and surveys such as (Zhou et al., 2021; Hoffman et al., 2019) emphasize the distinction between human-centered and model-centered explanation quality and the need for metrics that reflect different goals of interpretability. We extend these metrics by evaluating explanations across models rather than within a single model. We measure whether a CoT from one LRM generalizes behaviorally to others through cross-model consistency. This provides a complementary perspective to existing explanation-evaluation frameworks focused on human preference or model faithfulness.

## 5 DISCUSSION

In this work we have been motivated by the question, *"What is a good explanation?"* Unlike previous work that has focused on faithfulness or correctness, we have focused on the question of learnability: the notion that a good explanation should be effective at guiding a new student as the teacher intended. Taking advantage of the structure of LRMs as both producers and consumers of CoT, we have established a new framework for approaching this problem by measuring the generalization of CoT from one LRM to another.

Our work sets out a systematic way to conduct for an automated, quantitative evaluation of the generalizability of natural-language explanations, and we have demonstrated the use of our framework to measure a range of LRMs' ability to explain realistic tasks. We have shown that generalizability can be measured in terms of consistency of effects when transferring a CoT from one LRM to another. This consistency can be measured by removing explicit answers from the explanations, and measuring whether the other LRM arrives at the explained answer, right or wrong. We have also developed an ensembling method for generating highly-generalizable explanations.

Finally, our human study has validated that our consistency measure of generalization of chains-of-thought correlates with human preferences for better explanations. While our results here show some promise of LRM explanations at generalizing, the analyzed consistency metrics lay the groundwork for several interesting questions for future research, in particular raising the enticing possibility that LRMs could be trained to produce highly generalizable explanations. The framework of generalizable explanations also leads us to the question of whether the production of such explanations is itself a generalizable capability, and whether generalizable explanations produced by LRM can ultimately be used to help humans understand AI reasoning on subjects that extend human-produced training data, providing insights about knowledge beyond current human knowledge.

## REPRODUCIBILITY

All experiments were run on workstations with 141GB NVIDIA H200 SXM GPUs using the Hug-gingFace Transformers library (Wolf et al., 2020). The codes and the dataset produced during this work will be made publicly available after publication.

## ETHICS

Our work studies the generalizability of chain-of-thought (CoT) reasoning across different models and tasks. While CoT can improve performance and interpretability, its generalizability should be considered carefully. Reasoning patterns that transfer well in one setting may also reinforce shared mistakes in another, leading to consistent but incorrect outputs. In addition, reusing or combining CoTs across models may affect accuracy in ways that are not always predictable. These effects are particularly important to keep in mind in sensitive application areas, such as healthcare or law, where errors carry higher risks. We view this study as a step toward understanding both the benefits and limitations of CoT transfer. Future work should continue to explore when and how CoT generalizes reliably, and how to identify cases where it may not.

As for the user study we conducted, no personal information was collected during the user study experiments.

## USE OF LARGE LANGUAGE MODELS

We used LLMs to help with plotting and minor editing of paper text.

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

# A  SAMPLED DATA DISTRIBUTION

## A.1  MEDCALC BENCH

We randomly sampled 100 data points from the `MedCalc-Bench` with seed 42 for our experiments. To show that these points are representative, we calculated the default deterministic CoT model performance across the sampled and full dataset. Figure 8 shows that the trend of best to worst model performance remains the same across default and across empty variations.

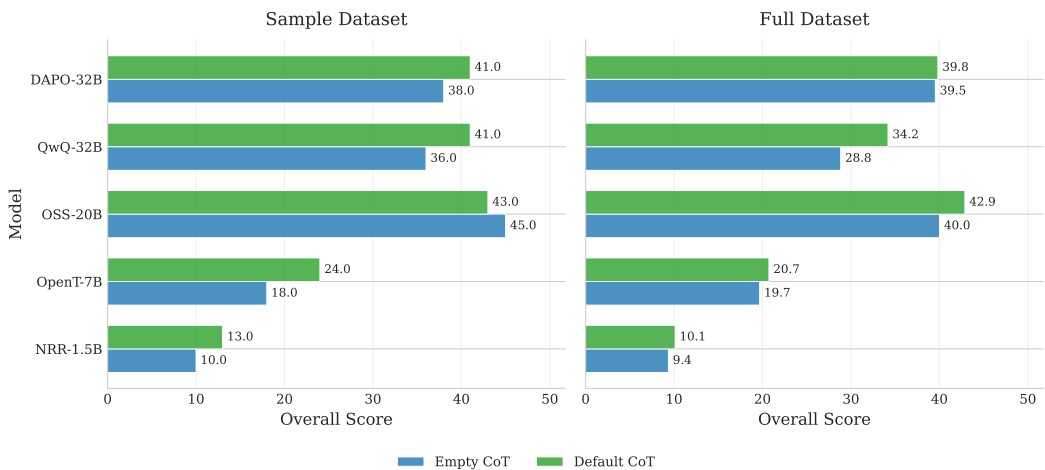

Figure 8: Model performance across all sampled and full data. The trends remain the same. Hence, the sample collected is a representative sample.

The model, `openai/gpt-oss-20b`, has three reasoning levels — low, medium, high. Based on the Medcalc benchmark, this model performs the best in the low level reasoning. Hence, for the rest of our experiments, we evaulated this model's CoT in low reasoning level effort.

# B  INSTRUCTION-INDUCTION

To create diverse and potentially complex instructions, we construct 12 new tasks in addition to the 24 tasks in the Instruction Induction Dataset (Honovich et al., 2022). The new tasks can be described as follows:

1. Reverse from middle: Locate the center point and reverse the left and right segments

2. Smallest Item Length: Find the shortest item and return its character count

3. Smallest even number square root: Identify the smallest even number and return its square root

4. Most vowel return consonant: Find the word with the most vowels and return its consonant count

5. Detect rhyme and rewrite: Detect rhyme schemes in poetry, then rewrite maintaining the same pattern.

6. Rank by Protein: Group foods into macronutrient categories and order by descending protein percentage

7. Translate to English: Recognize what language is being used and convert the main phrases into English

8. Square of Zodiac Animal: Find the zodiac animal in each list and output the square of its zodiac position

9. Alternate synonym antonym: Alternate between giving an antonym and synonym of the words in the sentence

10. Most consonant return vowel: Identify the word with the most consonants and return its vowel count

11. Identify fewest unique letters and return total letter count: Identify the word with the fewest unique letters and return its total letter count

12. First Word Alphabetically Return Reverse: Find the word that comes first alphabetically and return it in reverse

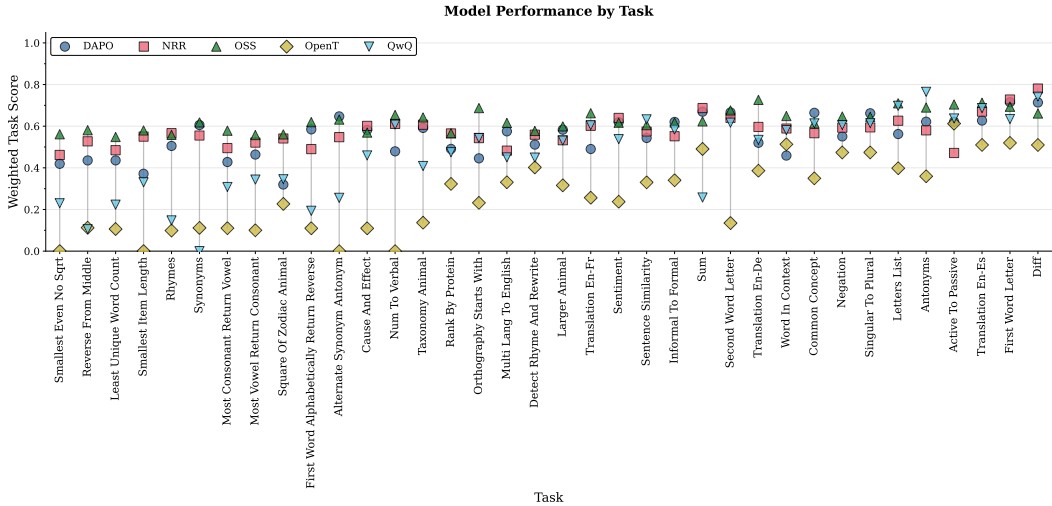

Figure 9: Model performance across all sampled data, including both the original instruction induction and newly introduced tasks.

## C    REMOVE ANSWERS

We prompt OpenAI's o4-mini (OpenAI, 2025) with the content presented in Listing 1.

```
f"""Task: Keep only the hints from the text and remove answer sentences.

Definition:
- A "hint/explanation sentence" provides guidance that helps someone
  think about the problem without giving the final solution.
- An "answer sentence" directly states the final answer, solution,
  result, or conclusion.

Instructions:
1. Keep every hint/explanation sentence exactly as written.
2. Remove all answer sentences and statements.
3. Preserve the original wording, order, and formatting of the
   remaining text.
4. Do not add, rephrase, or generate any new text beyond what is
   already in the original.
5. Output only the hints.

Original text:
{chain-of-thought}"""
```

Listing 1: User Prompt Content for Answer Removal

## D    ENSEMBLE STATS

The ensemble chain-of-thought method involves multiple models generating candidate sentences, after which judge or evaluator models select one candidate based on what they are least surprised by seeing, i.e., perplexity. The next sentence is then generated sequentially based on the context of

the question and the previous selected candidates, continuing this process iteratively. In Figure 10, we look at the distribution of candidates chosen from different pairs of generator models in various combinations of evaluated ensembles.

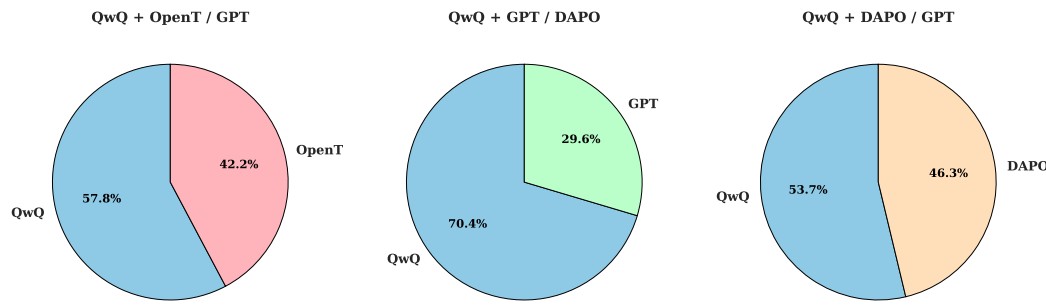

Figure 10: Proportion of selected candidate sentences from different generator models used to construct ensemble chain-of-thoughts across settings in MedCalc-Bench.

## E  MORE DETAILS ON USER STUDY

Fig. 11 shares a detailed overview of user study scores. We also share the detailed results of ilcoxon signed-rank test and paired t-test  Table 4.

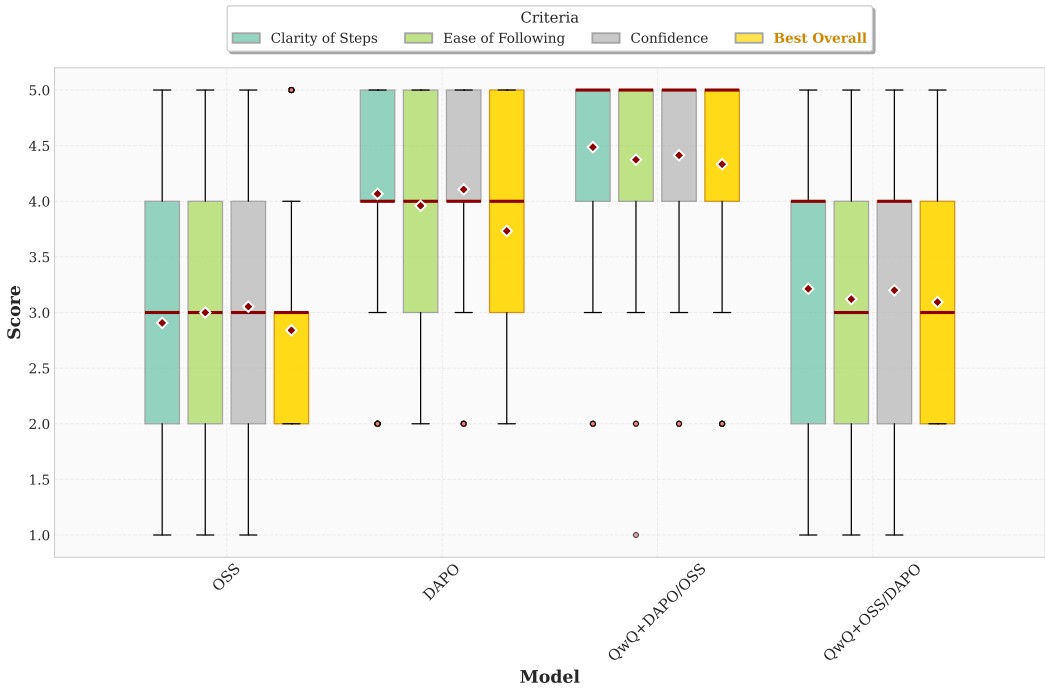

Figure 11: **User study results: Model performance comparison across evaluation criteria.** Box plots summarize the distribution of CoT evaluation scores for different models and model combinations. The three criteria are **Clarity of Steps** (leftmost box for each model), **Ease of Following** (second left box for each model), and **Confidence** (second right box for each model). Each box shows the median (red line), interquartile range (box boundaries), whiskers, and outliers (points), with red diamonds marking mean values. The **Best Overall** rankings (rightmost box for each model) report perceived understandability. Higher scores for all are better. The models are: OSS, DAPO, QwQ + DAPO/OSS and QwQ+OSS/DAPO.

Table 4: Pairwise Significance Matrix: Mean Differences (Wilcoxon Test)

| | OSS | DAPO | QwQ+DAPO/OSS | QwQ+OSS/DAPO |
|---|---|---|---|---|
| *Clarity of Steps* | | | | |
| OSS | – | $-1.16***$ | $-1.58***$ | $-0.31***$ |
| DAPO | | – | $-0.41***$ | $+0.85***$ |
| QwQ+DAPO/OSS | | | – | $+1.26***$ |
| QwQ+OSS/DAPO | | | | – |
| *Ease of Following* | | | | |
| OSS | – | $-0.96***$ | $-1.37***$ | $-0.12$ |
| DAPO | | – | $-0.41***$ | $+0.84***$ |
| QwQ+DAPO/OSS | | | – | $+1.25***$ |
| QwQ+OSS/DAPO | | | | – |
| *Confidence* | | | | |
| OSS | – | $-1.05***$ | $-1.36***$ | $-0.15*$ |
| DAPO | | – | $-0.31***$ | $+0.91***$ |
| QwQ+DAPO/OSS | | | – | $+1.21***$ |
| QwQ+OSS/DAPO | | | | – |
| *Best Overall* | | | | |
| OSS | – | $-0.89***$ | $-1.49***$ | $-0.25***$ |
| DAPO | | – | $-0.60***$ | $+0.64***$ |
| QwQ+DAPO/OSS | | | – | $+1.24***$ |
| QwQ+OSS/DAPO | | | | – |

Note: Values show mean differences (row model - column model). Green shading indicates row model rated lower (negative difference); red shading indicates row model rated higher (positive difference). Significance: ***$p < 0.001$, **$p < 0.01$, *$p < 0.05$. Gray indicates non-significant difference ($p > 0.05$). $n = 375$ pairs for all comparisons.

