# OpenReview forum: "Eliciting and evaluating generalizable explanations from large reasoning models"
_ICLR.cc/2026/Conference — Submitted to ICLR 2026_

### Official Review · Reviewer_yf2Q · 2025-10-29

**Soundness:** 2
**Presentation:** 3
**Contribution:** 1
**Rating:** 2
**Confidence:** 4

**Summary:**

This work is motivated by understanding if CoT explanations capture 'general patterns' about the underlying problem rather than patterns which are esoteric to each specific LRM. Experiments are performed substituting a model's native CoT with one of 3 CoT variants (no CoT, CoT from another model, or the "best" CoT picked from an ensemble of models) to evaluate whether the same CoT trace will produce the same final answer in different LRMs. A human user study is also conducted to determine that users prefer models whose CoT explanations are universal/consistent.

**Strengths:**

The paper is clearly written, the setup is easily understandable from the description, and Fig 1 is helpful.

**Weaknesses:**

My main issue with this work is the novelty/significance of the overall goal of this work and the actionable insights it generates. Even if users demonstrate a preference for consistency in CoT, I'm not sure why CoT traces have to be universal across models to be a sound explanation/reasoning trace (after all, different people can explain the same concept in very different ways). Furthermore, since the authors note that consistency is in fact quite high, the result is not surprising nor does it generate interesting directions for future research. The result of the user study (users prefer models with consistent CoT) also is not surprising. In order for me to raise my score, I would need to see an actionable, surprising result (e.g., a systematic failure of LRMs with certain types of CoT).

Small note: The PDF document attached in OpenReview has a different title than the OpenReview submission.

**Questions:**

How do the authors define 'generalizable'? In this context I would probably say the work is testing whether CoT traces are 'universal' or 'consistent'; 'generalizability' is a property of a model.

---

> ### Author Response · Authors · 2025-11-21
> **Rebuttal**
>
> Thank you for your thoughtful and constructive feedback. We appreciate your positive remarks on the clarity of our work, and we understand your concerns regarding the novelty and significance of our contributions. In the revision, we will make these aspects more explicit and we address each of your concerns in detail below:
>
> **Title of the paper:** Thank you for catching this discrepancy. To better reflect our discussion with the reviewers and clarify the paper’s motivation, we are updating the title to “Do explanations generalize across large reasoning models?” We will contact the AC to update the title on the OpenReview page.
>
> **Definition of “generalizable”:** In our work, we use “generalizable” to describe the degree to which various models reach the same final answer when conditioned on the same chain-of-thought (CoT) setting. While generalizability typically refers to a model’s ability to transfer performance on unseen domains, we study a complementary notion, i.e., generalizability across models given a shared CoT setting. This includes, for example, transferring a CoT generated by one model to others and observing whether these models reach the same conclusion.
>
> As you note, consistency is the more precise, operational measure of this phenomenon, and we do use consistency as our metric throughout the paper. “Generalizability” is intended as the conceptual framing (i.e., whether the same conclusion is reached amongst models under the same CoT setting), and we will clarify this distinction to avoid ambiguity.
>
> **Novelty and actionable insight:** We agree that models and humans may reasonably provide different explanations for the same concept. Our focus is not on enforcing identical explanations, but on evaluating **whether a given explanation reliably leads different models to the same conclusion**. This perspective highlights an underexplored dimension of CoT research: whether a reasoning trace functions as a stable and interpretable basis for different models to reach the same conclusion.
>
> Although overall consistency is high, our results reveal nontrivial and actionable failures. In particular, GPT-OSS-20B demonstrates relatively high accuracy (Table 1, Transferred CoT (OSS)) but notably lower consistency than other CoT sources (e.g., Transferred CoT (DAPO) and some ensemble CoTs in Figure 5). This inconsistency correlates to lower perceived explanation quality, as reflected in the significantly lower human ranking of OSS CoT (orange labels in Figure 4).
>
> This gap between accuracy and cross-model consistency is precisely the type of phenomenon that suggests actionable direction for future research. For example, RL-based training pipelines might explicitly incorporate cross-model consistency (or generalizability across models) as a reward signal, encouraging models to produce CoTs that are robust and interpretable to users and other reasoning models.
>
> We hope that we were able to address your concerns and improve the paper. Please let us know if you have any remaining concerns and/or suggestions for improvement.

---

### Official Review · Reviewer_5iWP · 2025-10-30

**Soundness:** 3
**Presentation:** 2
**Contribution:** 3
**Rating:** 6
**Confidence:** 3

**Summary:**

This paper studies whether explanations from large reasoning models generalize: whether they induce the same behavior when provided to other LRMs. They use accuracy and consistency to evaluate. When evaluating, they use an LLM to remove the answer from the reasoning. They also evaluate empty CoT (no "thinking token"), default CoT (the thinking that is done by the same model), transfer CoT (patching one model's CoT to another model), and ensembled thoughts (have generator models generate multiple reasonings, and an evaluator model to choose different steps from differemt model generations to put together into one).  They find that CoT explanations do generalize and make outputs more consistent, even for ones that make wrong answers, and the ones that generalize (produce more consistent answers) are rated higher by users too.

**Strengths:**

1. The paper is well motivated, studying one aspect of a good explanation: one that should guide the user to the same conclusion. This angle is important and novel.
2. The paper shows that using the same explanation does induce more consistent behavior than just having the model generate on its own, which is evidence of how the explanations generalize. This is a new way of testing explanations.

**Weaknesses:**

1. The human study size is very small. Also since the participants see multiple examples, it should be paired tests instead of independent t-tests. And there is a debate between if we should treat likert scales as parametric or non-parametric. It might be good to add Wilcoxon signed-rank test as well as paired t-test.
2. Figure 5 shows that if we use the same CoT on the same model that created the CoT, then it produces more consistent answers than if we use a different model. But the paper did not show how if we just sample from the model (which might give different CoT) whether the result will be different. It is thus hard to interpret what this "CoT creator involved" results should be interpreted.
3. Some typos:

line 047 a problems -> a problem

the last line of page 1: MedCalc Bench. (add period)

line 093: duplicated "they"

**Questions:**

1. (W1) Could you add Wilconxon signed-rank test and paired t-test to the human study?
2. (W2) Could you add results of how a model agrees with itself just by sampline different outputs from the model and test self-consistency? This way we can show that if using CoT from the same model actuall improves from just sampling from the same model.

---

> ### Author Response · Authors · 2025-11-21
> **Rebuttal**
>
> Thank you for your thoughtful feedback and your kind words about the motivation and the novelty of our work in testing explanations. We address your questions and suggestions below and in our revision.
>
> **Q1 & W1. Add Wilcoxon and paired t-test.** Thank you for the suggestion. We have added this test, which helped strengthen our findings. We found that the results are consistent with the independent t-test with Bonferroni correction. We have added this test result in lines 374-376 and added the detailed statistics in Appendix E.
>
> **Q2. and W2 Self consistency and cross-model consistency.** We added Figure 7 to illustrate the models’ self-consistency, which ranges from 20% to about 50% for respective models in question. This differs from cross-model consistency of models sampling their responses. In setting B of figure 1, we state two versions: 2.1. The deterministic version where we use the model's deterministic decoding for generating the CoT and 2.2. The sampling version where we sample from the same model. The results of the sampling consistency across models (i.e., if we sample from the same model and keep this setting for the rest of the models, do these models come to the same conclusion) are shared as orange bar results in figures such as figure 5. These are comparable to the deterministic default version which is labeled as “default”. Essentially, in sampling and default versions, we are looking at whether various models reach to the same conclusion inherently. This shows cross-model consistency of their own settings.
>
> **W2. Creator involved clarification.** Figure 6 (previously known Figure 5) looks at how the average pairwise consistency rate changes if one of the pairs included the source model (CoT creator involved) or both models in the pair are target models (i.e., they were not used in the creation of CoT). This essentially tries to show to what extent similarity of results still occurs if one of them is the source vs none of them are the source.
>
> **Typos.** Thank you for catching these typos. We revised the paper to resolve these typos.
>
> We hope that we were able to address your concerns and improve the paper. Please let us know if you have any remaining concerns and/or suggestions for improvement.

---

### Official Review · Reviewer_APnE · 2025-10-31

**Soundness:** 2
**Presentation:** 3
**Contribution:** 2
**Rating:** 4
**Confidence:** 3

**Summary:**

This paper proposes a method for evaluating if and how well chain-of-thought (CoT) reasoning traces can serve as general explanations. The basic idea is that, if a CoT reasoning trace is not just sequence of model-specific "thinking token"-like outputs that allow the model to spend more computation on the problem, but a series of generally applicable reasoning steps, then providing such CoT reasoning traces should be useful for (a) other models and (b) human users.
The paper operationalizes this idea in two main ways:
1. Providing CoT reasoning trace z produced by model A as input to models B_1, ..., B_n and then evaluating if this improves task accuracy with respect to ground truth, as well as consistency, i.e., does reasoning trace z elicit the same answer in models B_1, ..., B_n?
2. Collecting human ratings of LM-generated CoT reasoning traces. The paper finds a moderate positive correlation between human ratings and the accuracy and consistency scores measured in step 1.

In addition to these two main contributions, the paper also devises a method for aggregating multiple reasoning traces from different models and shows that such an ensemble yields the best explanations under the considered evaluation metrics.

**Strengths:**

The idea of evaluating the explanatory function of CoT reasoning traces by providing them to other models and humans is interesting and, to my knowledge, novel. The evaluating framework, which includes various combinations of LMs as well as human ratings, is well-designed.

**Weaknesses:**

While I think the idea and overall framework is a valuable contribution, the scale of the presented experiments does not lead to convincing conclusions.

1. Experiments are only performed on one dataset with 100 samples. I realize that human evaluation is costly, but at least for the LLM-based evaluation, larger-scale experiments both in terms of datasets/tasks and sample sizes should lead to more robust findings.

For example, the observed positive correlations between consistency/accuracy and human user ratings (Figure 3, top two figures) is based on four datapoints. It appears that excluding a single datapoint, namely the QwQ+DAPO/OSS result, would flip the sign of correlation, so I find myself having to question the robustness of this result.

2. There is a mismatch between the model and human evaluation. The model evaluation measures accuracy with respect to ground truth (measuring whether the explanation helps models perform the task better) and consistency (measuring whether explanation elicits possibly wrong, but consistent answers in different models). In contrast, the human annotators are asked to rate aspects like "clarity" and "confidence" and then the paper correlates these ratings with model evaluation metrics. Simply using the same metrics for humans and models would allow for a direct comparison. If having humans actually perform the task is too difficult, I would argue that (related to point 1.) experiment should also included a task that humans can reasonably do.

3. The analysis of model results limited. When transferring a CoT reasoning trace from one model to another, there are several possible cases which, I believe, should be analyzed to better understand when and how CoT transfer works. For example:
- Model A predicts the wrong answer without CoT but correct answer with CoT z_A. Model B predicts wrong answer without CoT and with its own CoT z_B. When transferring CoT z_A from A to B, model B now predicts the correct answer with CoT z_A. This would be a case in which CoT z_A is effective and in this sense z_A is a good CoT that "generalizes".
- Model A predicts the wrong answer with CoT z_A, Model B predicts the correct answer without Cot and with its own Cot z_B. There is now a conflict between model B's tendency to predict the correct answer and transferred CoT that goes against that tendency. It's less clear what outcome we want here, but if z_A flips model B's prediction to a wrong answer, at least we could say that z_A is "convincing".
There are several more of these possible combinations, and I would have expected some analysis along these lines to better understand the proposed method. However, related ot 1., the small sample size likely does not allow such an analysis.

**Questions:**

- lines 117-220 are confusing: what is the difference between an "eval" model and a "test" model? Why is the explanation z given to a "to a different LRM leval and a set of other test models L = {li, lj ...lq }" but then, if I understand "We then produce an answer given an LRM_leval" correctly, the "other test models" are not used? Looking at Eq (1), it appears that the test models are used for the consistency metric G, but this is not explictly explained.

- Eq (1): does the index j in the equation for G refer to the j in the definition of L on line 117 or this a typo for the summation index i? (Guessing the latter, since i doesn't appear anywhere else...)

---

> ### Author Response · Authors · 2025-11-21
> **Rebuttal**
>
> Thank you for your detailed feedback. We appreciate your recognition of the novelty of our work and well-designed evaluation framework. We address your questions and suggestions below and in our revision.
>
> **W1. Scale and Robustness of experiments.** To strengthen the robustness of our findings, we expanded our experiments to an additional benchmark: Instruction Induction. We now report results on this dataset alongside the original one. The new experiments confirm the trends observed in our initial results, i.e., the ensemble-based approach exhibits greater stability and CoT-transfer effects persist (though with variation across model sources). We updated Table 3 and Figure 5 to include accuracy and consistency results for the additional benchmark.
>
> **W2. Mismatch between model and human evaluation.** We agree that asking human annotators to perform the full underlying tasks is not reasonable, which is why we evaluated different human-interpretable dimensions (clarity, ease of following, confidence). These dimensions capture whether the explanation is convincing or helpful for humans, in contrast to model-based accuracy and consistency. We have clarified this distinction in the revision and made explicit why we do not ask humans to solve the original tasks directly.
>
> **W3. Limited analysis of model outcomes.** We appreciate this suggestion and have expanded our analysis accordingly. In Figure 3, we now report a breakdown of outcomes comparing CoT and baseline empty CoT reasoning across several scenarios, including cases where (a) an incorrect model prediction becomes correct after CoT transfer, (b) a correct prediction is perturbed, and (c) different forms of agreement or disagreement emerge across models. This breakdown addresses the kinds of cross-model dynamics you highlighted. We note that the sample size still limits the granularity of certain analyses, but the new results provide substantially more insight.
>
> **Q1. Difference between eval and test models.** Thank you for pointing out this confusion; we have revised the text for clarity. In our setup, $l_{eval}$ is one of the models in the evaluation set $L$. For accuracy, we measure only whether $l_{eval}$ answers correctly. For consistency $G$, we compute pairwise agreement across all models in $L$. The revision clarifies that the “other test models” contribute to the consistency calculation, even though the accuracy metric focuses solely on $l_{eval}$.
>
> **Q2. Clarification of Equation 1.** Thank you for catching the index consistency. We have corrected the notation by writing the summation over $l \in L$, making it explicit that all models in $L$ are used for pairwise consistency computation.
>
> We hope that we were able to address your concerns and improve the paper. Please let us know if you have any remaining concerns and/or suggestions for improvement.

---

### Official Review · Reviewer_pVAd · 2025-11-01

**Soundness:** 2
**Presentation:** 2
**Contribution:** 3
**Rating:** 2
**Confidence:** 3

**Summary:**

The paper investigates whether CoT explanations generated by LRMs are generalizable. The author define generalizability as cross-model consistency whether an explanation from a source LRM induces different receiver LRMs to produce more consistent answers. Using this evaluation framework on MedCalc Benchmark, the authors find that CoT-based hints, which is the explanation with answers removed, significantly increase the pairwise consistency of receiver models' answers compared to no-explanation baseline.

**Strengths:**

1. The paper is well-written and easy to follow.
2. The paper tackles the critical question of whether CoT explanations capture general which is a significant question for the community.
3. The experimental validation is thorough employing diverse set of LRMs

**Weaknesses:**

1. The paper title and pdf title is different.
2. The main text of the figure and table needs to provide guidance on how to interpret the results in the table (e.g., what the takeaway is) (for figure 2,5, table 3). Also the readability of the figures are poor (for figure 2,3,4).
3. The experiment is only held in one medical benchmark dataset which lacks generalizability. The author should mention 'medical domain-specific' in the title or do additional experiment across different domains.
4. The framework measures consistency of the final answers but not their correctness. There should be an analysis of correctness since it might consistent on incorrect answer.
5. The paper positions itself in the LRM reasoning space but largely bypasses the decades of xai research on explanation evaluation.
6. The paper equates high consistency with generalization. This needs formal justification or empirical validation.

**Questions:**

Look at the weaknesses

---

> ### Author Response · Authors · 2025-11-21
> **Rebuttal**
>
> Thank you for your detailed feedback. We are glad that you found our work to be well-written and tackling a significant question for the community. We address your questions and suggestions below and in our revision.
>
> **W1. Title inconsistency.** Thank you for catching this. To better reflect our discussion with the reviewers and clarify the paper’s motivation, we are updating the title to “Do explanations generalize across large reasoning models?” We will contact the AC to update the title on the OpenReview page.
>
> **W2. Need for clearer interpretation guidance + figure readability.** We appreciate this feedback. We will revise the paper to provide explicit interpretation guidance for the relevant figures and tables and improve figure readability (larger fonts and relocation of dense user-study plots to the appendix). Below is a summary of the additions.
>
> - *Figure 2:* We measure how often a Chain-of-Thought (CoT) leads models to converge to the same answer even when the predicted answer is not mentioned in the CoT (i.e., when the CoT mostly provides partial hints or reasoning). The left panel breaks down these “same answer” cases into convergence on the same correct versus the same incorrect answers. The right panel shows the proportion of consistently incorrect answers across CoT types. These results demonstrate that CoTs can systematically steer model reasoning, including toward the same conclusion, which indicates that CoT can exert a generalizing influence on model behavior even when the reasoning they provide can be incorrect.
> - *Figure 6 (previously known as Figure 5):* We compare the consistency rate for model pairs where at least one model contributed to the source CoT vs pairs where neither did. The takeaway is that consistency persists both within-source and out-of-source model pairs, which supports the existence of generalizable reasoning traces.
> - *Table 3:* We report accuracy for each CoT type. The takeaway is that models with weaker baseline accuracy can reduce the accuracy of other models when their CoTs are transferred even if the other model inherently performs better at their own baseline.
> - *Figures 2, 3, and 5 (previously known as 4):* We increased the font size and moved additional user-study distribution plots to the appendix.
>
> **W3. Generalization beyond the medical domain.** Thank you for emphasizing this point. To improve generalizability, we have included an additional non-medical dataset: Instruction Induction. The expanded results (updated Table 3 and Figure 5) show that the major trends hold across domains: ensemble CoTs remain more stable, and CoT transfer effects persist with model-dependent variation.
>
> **W4. Measure correctness in addition to consistency.**  We agree that consistency alone does not guarantee correctness, and we directly address this in our analysis. Figure 2 breaks down consistent outputs into matching-correct and matching-incorrect cases, and also reports the overall rate of consistent-but-wrong answers across thought settings. This allows us to see not only whether models align, but whether they can align when the reasoning can be right or wrong. To deepen this analysis, Figure 3 examines how transferring CoTs changes prediction outcomes in the models themselves across various scenarios (e.g. gets initially wrong in empty CoT baseline to correct answer, initially correct to wrong answer). This provides a detailed view of when CoTs help models correct errors, when they probate mistakes, and when they preserve the correct reasoning.
>
> **W5. XAI explanation evaluation literature.** Thank you for pointing this out. We agree that connecting our work more explicitly to the broader explanation-evaluation literature will improve the paper. In the revision, we add more related papers in the “evaluating natural-language explanations” section to dedicate discussion highlighting how our approach relates to prior work in explanation evaluation within XAI. We also clarify that our focus is on a specific aspect of explanation behavior, i.e., how explanations transfer across models, which complements previous human-centered explanation evaluation frameworks.
>
> **W6. Consistency vs Generalization.** Thank you for raising this point. We have clarified our paper title, abstract, and intro to be very clear about the type of generalization we are studying: “whether explanations produced by one LRM induce the same behavior when given to other LRMs.” and measure consistency and accuracy (Eq 1) to evaluate this generalization. We believe this notion of generalization across models is important and have clarified the discussion around it to avoid implying anything broader that the term “generalize” may suggest.
>
> We hope that we were able to address your concerns and improve the paper. Please let us know if you have any remaining concerns and/or suggestions for improvement.

---

### Author Response · Authors · 2025-11-21
**Overall Rebuttal Comment**

We are grateful to the reviewers for their thoughtful and constructive feedback. We are glad that the reviewers have found our paper well-written [pVAd,yf2Q], well-motivated [pVAd, 5iWP29], and novel [APnE31, 5iWP29]. We have updated our submission to incorporate their feedback, which we believe has strengthened our submission. Below we summarize and address the common concerns:

- **Title of the paper** [pVAd, APnE31, yf2Q29]: To better reflect our discussion with the reviewers and clarify the paper’s motivation, we are updating the title to “Do explanations generalize across large reasoning models?” We will contact the AC to update the title on the OpenReview page.

- **Additional benchmark** [pVAd, APnE31]: To improve the robustness of our findings, we expanded our experiments to an additional benchmark: Instruction Induction. We now report results on this dataset alongside the original one. The new experiments confirm the trends observed in our initial results, i.e., the ensemble-based approach exhibits greater stability and CoT-transfer effects persist (though with variation across model sources). We updated Table 3 and Figure 5 to include accuracy and consistency results for the additional benchmark. More information about this benchmark can be found in Appendix B.

- **Clarification on generalizability and its novelty** [pVAd, yf2Q29]:  Our focus is not on enforcing identical explanations, but on evaluating whether a given explanation reliably leads different models to the same conclusion. This perspective highlights an underexplored dimension of CoT research: whether a reasoning trace functions as a stable and interpretable basis for different models to reach the same conclusion.

We invite the reviewers to examine the updated submission and welcome any additional questions, comments, or suggestions for further improvement. If you wish to see the tracked changes, you can view the anonymized pdf linked here: https://anonymous.4open.science/r/rebuttal-2550/diff.pdf.  We also respond to each individual reviewer in separate responses.

---

### Author Response · Authors · 2025-12-03
**Acknowledgment to Reviewers**

We extend our sincere gratitude to the reviewers for their thoughtful feedback and for helping us substantially strengthen the paper. In our revision, we have fully addressed all major concerns:
* **Robustness & scale:** We added a second benchmark (Instruction Induction), doubling task diversity. All core trends replicate across datasets, reinforcing our main claims (updated Table 3, Fig. 5).
* **Correctness vs. consistency:** We added a detailed CoT-transfer outcome analysis (Fig. 3), directly addressing requests for deeper interpretability and revealing when CoTs correct or propagate errors.
* **Human-study statistics:** We added paired t-tests and Wilcoxon signed-rank tests (App. E), confirming the statistical significance of user preferences for more generalizable CoTs.
* **Novelty & significance:** To our knowledge, this is the first systematic study of cross-model generalization of reasoning traces. Our results uncover an actionable divergence between accuracy and cross-model reliability (e.g., GPT-OSS-20B), suggesting a concrete direction for future work: training models to produce reasoning traces that transfer robustly across architectures.

We also revised key definitions, expanded XAI-related-work connections, clarified eval/test roles, fixed Eq. 1, and improved figure readability throughout. We respectfully ask the AC to evaluate the revised manuscript in light of these substantial improvements. We believe the strengthened version offers clear novelty, actionable insights, and a valuable new perspective on the transferability of CoT explanations. We are encouraged that reviewers found the paper well-written [pVAd, yf2Q], well-motivated [pVAd, 5iWP29], and novel [APnE31, 5iWP29].

Sincerely,

The Authors

---

### Meta-Review · Area_Chair_Yhfo · 2025-12-28

**Summary:**

This paper investigates whether explanations can generalize across reasoning models. The main mechanism of this is to transfer CoTs between models or ensemble a CoT from multiple models, then use that as a CoT in each model. The goal is to determine whether an explanation is a "real" explanation of the phenomenon at hand or "patterns which are esoteric to the LRM".  The paper draws correlation between generalizability of explanations between models and human judgments of the quality of those explanations.

Strengths:

- Clearly written paper

- Interesting study of transferring CoTs between models

Weaknesses:

I see two fundamental flaws with this study.

First, as yf2Q points out, there can be different ways to get to the same answer, so being able to answer from an explanation from another model is not sufficient for determining whether an explanation "capture[s] general patterns about the underlying problem rather than patterns which are esoteric to the LRM".

Second, this paper fails to address label leakage, which is a major issue with the methodology of transferring an explanation and attempting to predict the answer.  See Hase et al. "Leakage-Adjusted Simulatability" https://arxiv.org/abs/2010.04119 for a paper on label leakage in explanations. An explanation *cannot* be viewed as independent of the answer.  Instead, it should be viewed as communicating some level of certainty about that answer ("The answer is A": very high certainty. Gibberish: no certainty. Other explanations: in the middle.)

As a result, I'm not sure we can actually conclude anything meaningful about explanation quality from transfer. A greater score in Figure 2 may just indicate that the correct explanation is more strongly "hinted" in the response. The paper only discusses removing explicit answer declarations, but otherwise doesn't address this point, yet it's a critical weakness for the validity.

In spite of these flaws, generalizability as measured in this paper *could* still be meaningful based on the correlation with human rankings: even if it is a flawed metric in and of itself, if it correlates with human judgments, it's useful. However, I am not strongly convinced by Figure 4, sharing the weaknesses mentioned by APnE. The number of data points is simply too few to establish a strong pattern, especially as models that are poor on both consistency and user ratings may just produce poor quality explanations in general.  That is, there is a shared feature of overall explanation quality, and generalization isn't really established by these results.

**Reviewer Concerns:**

- Lack of generality due to narrowness of benchmark and number of samples (pVAd, APnE): somewhat addressed by a new domain, but remains

- Lack of evidence for trends in figures (APnE): somewhat addressed

- Underbaked analysis of cross-model CoT transfer (APnE): fundamentally remains

- Premise of the work is flawed (yf2Q): there can be different ways to get to the same answer, so being able to answer from an explanation from another model is not sufficient for determining whether an explanation "capture[s] general patterns about the underlying problem rather than patterns which are esoteric to the LRM": fundamentally remains

- Lack of actionable insights from the work (yf2Q): fundamentally remains

- "The paper positions itself in the LRM reasoning space but largely bypasses the decades of xai research on explanation evaluation." (pVAd) Although the reviewer didn't mention an explicit line of literature here, the papers I cited give an example of work that should've been engaged with. This concern fundamentally remains.

**Reviewer Scores:**

Adding an additional domain could help convince pVAd. Reviewer APnE has other issues with the work. I don't think yf2Q would be convinced.

---

### Decision · Program_Chairs · 2026-01-26

Reject